# *Cladosporium* Species: The Predominant Species Present on Raspberries from the U.K. and Spain and Their Ability to Cause Skin and Stigmata Infections

Lauren Helen Farwell [1,2,*], Greg Deakin [1], Adrian Lee Harris [1], Georgina Fagg [1], Thomas Passey [1], Carol Verheecke-Vaessen [2], Naresh Magan [2] and Xiangming Xu [1,*]

1. NIAB—New Road, East Malling, West Malling ME19 6BJ, UK
2. Applied Mycology Group, Environment and AgriFood Theme, Cranfield University, College Road, Bedford MK43 0AL, UK
* Correspondence: lauren.farwell@niab.com or lauren.farwell@cranfield.ac.uk (L.H.F.); xiangming.xu@niab.com (X.X.)

**Abstract:** Raspberry (*Rosales*: *Rosaceae*) production in the U.K. has moved rapidly in the last 10 years to under polythene, combined with a reduced availability of broad-spectrum fungicides. Hence, the incidence of previously less prevalent diseases, such as *Cladosporium* (*Capnodiales*: *Cladosporiaceae*), has largely increased. This study aimed to identify the predominant *Cladosporium* species on raspberry and to understand the nature of its infection on raspberry fruit. Raspberries were collected from farms across the U.K. and Spain and incubated; fungal isolates were then isolated from typical *Cladosporium* lesions and identified to the species level based on the sequences of the trans elongation factor α and actin genes. *Cladosporium cladosporioides* (Fres) de Vries was confirmed as the predominant species responsible for infecting raspberry fruit close to harvest on fruit from the U.K. and Spain, being present on 41.5% of U.K. fruit and 84.6% of Spanish fruit. Raspberries were subsequently inoculated at different developmental stages with *C. cladosporioides* isolates to determine the susceptibility to *Cladosporium* skin lesions and stigmata infections in relation to the developmental stage. Only the ripening and ripe raspberries were susceptible to *Cladosporium*, resulting in skin lesions. *Cladosporium* can colonise the stigmata of raspberries earlier in fruit development and future research is required to determine if such stigmata infections could cause subsequent skin lesion infections. This study has provided the necessary epidemiological information to develop effective management measures against the *Cladosporium* species.

**Keywords:** *Rubus ideaus*; epidemiology; inoculation; stigma; phylogenetics; susceptibility; ordinal regression





## 1. Introduction

Raspberry (*Rubus ideaus*; (*Rosales: Rosaceae*)) is an economically important soft fruit crop in the U.K. worth approximately £133 M/$171 M in 2020 [1], mostly grown under protection. Raspberries are delicate fruits susceptible to various fungal diseases, especially grey mould caused by *Botrytis cinerea* (*Helotiales: Sclerotiniaceae*). Based on recent epidemiological knowledge of raspberry grey mould on fruits under protection [2], a management strategy based on post-harvest cool-chain management without fungicides was developed and able to control grey mould effectively [3].

Recently, some U.K. raspberry growers reported significant losses in yield due to fruit rots caused by the *Cladosporium* (*Capnodiales: Cladosporiaceae*) species. Currently, there is insufficient epidemiological knowledge on raspberry fruit rot caused by the *Cladosporium* species to develop management strategies. One recent study in the U.S.A. examined the relationship between the *Cladosporium* species and *Drosophila suzukii* (*Diptera, Drosophilidae*) on raspberries; *C. cladosporioides*, *C. anthropophilum* Sandoval-Denis, Gené and Wiederhold and *C. pseudocladosporioides* Bensch, Crous and Braun were found to be the predominant

species [4]. In the U.K., it is unknown which predominant *Cladosporium* species are causing raspberry fruit rot. Previous studies of *Cladosporium* raspberry rot have focused on post-harvest disease development [5]. For example, *Cladosporium* was found to be prevalent at one farm in Kent, U.K., where 55% of raspberries exhibited *Cladosporium* symptoms post-harvest [3]; however, the origin of this high post-harvest incidence of *Cladosporium* was unclear. In the U.S.A., Swett et al. [4] showed that *Cladosporium* was present on fruit pre-harvest and on under-ripe raspberries when harvested and then incubated for 4 days.

If *Cladosporium* can infect raspberry fruits pre-harvest, the level of infection may depend critically on fruit age, as was found for *Botrytis* [6] and powdery mildew [7] on strawberry. *Cladosporium cladosporioides* was reported to cause blossom blight of strawberries in Korea [8] and the U.S.A. [9]. There is no knowledge of whether the *Cladosporium* species can similarly infect raspberry flower tissues, such as the stigmata, and become established on the fruit surface pre-harvest, resulting in yield losses. There is, therefore, an urgent need to understand the predominant *Cladosporium* species on raspberries and the susceptibility of raspberry fruit at different developmental stages to infection by *Cladosporium*. This knowledge will facilitate better development of control strategies to minimise the impact of this fungal infection at both the pre- and post-harvest stage.

Thus, the objectives of this study were to (a) determine which were the predominant *Cladosporium* species present on U.K. and Spanish raspberry fruit; (b) investigate fruit susceptibility to *Cladosporium* in relation to fruit developmental stages; and (c) identify whether stigmata are susceptible to *Cladosporium* infection.

## 2. Materials and Methods

### 2.1. Determining Cladosporium Species Present on Raspberry Fruit

#### 2.1.1. Isolate Collection

Raspberries were sent directly by U.K. growers or purchased from supermarkets in punnets (including fruit originating from Spain). The raspberries were incubated in polythene-bagged trays to maintain high humidity in ambient room conditions for four days to accelerate fungal growth. Fruit with typical *Cladosporium* lesions were dissected and the diseased parts were directly plated onto potato dextrose agar (PDA; Oxoid, Basingstoke, U.K., CM0139) in 9 cm Petri dishes (supplemented with 9.3 g per 100 mL NaCl to inhibit bacterial growth). The fungal outgrowth that resembled the *Cladosporium* species was then sub-cultured onto fresh PDA medium to obtain pure cultures.

#### 2.1.2. DNA Extraction, Amplification and Sequencing

DNA from those candidate *Cladosporium* isolates were extracted using a rapid fungal DNA extraction method with Sigma extraction and dilution buffers [10]. An approximately 1 mm × 1 mm piece of mycelium was taken from each isolate and placed in 30 µL of extraction buffer (Sigma-Aldrich, Gillingham, UK, E7526) and heated to 95 °C for 5 min. Then, 30 µL of dilution buffer (Sigma-Aldrich, Gillingham, UK, D5688) was added, followed by 50 µL of sterile water. The final product was stored at −20 °C for subsequent downstream processes. Once the DNA was extracted, PCR was performed on the trans elongation factor α (TEF1α) (primer pair: EF1-728F and EF-2R; Table S1) and actin (ACT) (primer pair: ACT-512F and ACT-783R; Table S1) regions, as these have been shown to be accurate in identifying *Cladosporium* at the species level [11]. The PCR cycle for TEF1α was set as follows: 95 °C for 8 min, 40 cycles of 95 °C for 15 s, 55 °C for 20 s, 72 °C for 1 min and, finally, 72 °C for 5 min. For the ACT gene, it was set as follows: 94 °C for 5 min, 45 cycles of 94 °C for 45 s, 52 °C for 30 s, 72 °C for 90 s and, finally, 72 °C for 6 min, as described by Swett et al. [4]. The PCR products were run on a 1.5% agarose gel for 45 min at 100 V with a 1 kb base pair ladder. If strong bands were seen, the products were diluted 1:10 and then sent to Eurofins (Ebersberg, Germany) for Sanger sequencing. The chromatograms were trimmed to remove poor end regions, and any sequences with signs of contamination were removed. For each isolate, the forward and reverse reads were combined for each gene to create a consensus sequence using Geneious version 2019.2.1 (Auckland, New Zealand).

Individual consensus sequences were then run through the nrBLAST database using the megaBLAST version 2.12.0+ (Bethesda, MD, USA) programme for species identification based on the e-value (<10–80) and the percentage identity (>95%). If multiple species were found with similar probability, the species identity was resolved as ambiguous. In total, 54 isolates were sequenced for both ACT and TEF1α regions.

2.1.3. Phylogenetic Analysis

Sequences from the BLAST searches with high similarity (e-value < 10–80, percentage identity > 95%) were added to our query sequences in the phylogenetic analyses as reference sequences (Table S2). As some of the Centraalbureau voor Schimmelcultures (Westerdijk Institute, Utrecht, The Netherlands) *C. fusiforme* isolates were missing the 3′ locus of the TEF1α gene, they were excluded in the TEF1α tree. In total, 73 sequences were used for the ACT phylogenetic tree at 215 base pairs long and 71 isolates for the TEF1α tree at 508 base pairs long, with *Cercospora beticola* (CBS 116456) as the outgroup in both trees. Sequences were aligned separately for the ACT and TEF1α genes using the online version of MAFFT version 7 (Osaka, Japan) (https://mafft.cbrc.jp/alignment/server/index.html (accessed on: 27 October 2022); [12]) with the E-INS-I model. Alignments were visually inspected, and sequences were trimmed to remove the primers using the software MEGA version X (Philadelphia, PA, USA) [13]. MEGA was also used to build maximum likelihood trees with 1000 bootstrap replicates for the two gene regions, in which a Kimura 2-parameter evolutionary model [14] was used for the ACT and TEF1α regions.

*2.2. Susceptibility of Fruit to Cladosporium Skin Lesions*

2.2.1. Experimental Design and Treatments

Inoculation experiments were conducted in an experimental raspberry plantation in a polytunnel in late summer 2019 and 2020 at NIAB, East Malling in Kent, U.K., (Longitude: 51.288350, Latitude: 0.454766). Potted plants of one proprietary raspberry variety (one of the most popular commercial varieties grown in the UK on which *Cladosporium* was previously reported by U.K. growers) were used and grown with an industry-standard fertigation regime for this specific cultivar. All three *C. cladosporioides* isolates used for inoculations were from U.K. farms and included in the phylogenetic analyses.

Fruits were classified into three developmental stages (green, ripening and ripe berries (Figure 1)) at the time of inoculation. Inoculations were conducted three times each year. On each plant, all fruits from one or more randomly selected branches with the widest range of fruit developmental stages were inoculated on each inoculation occasion unless otherwise specified. Any visible rotting fruits were removed prior to inoculation.

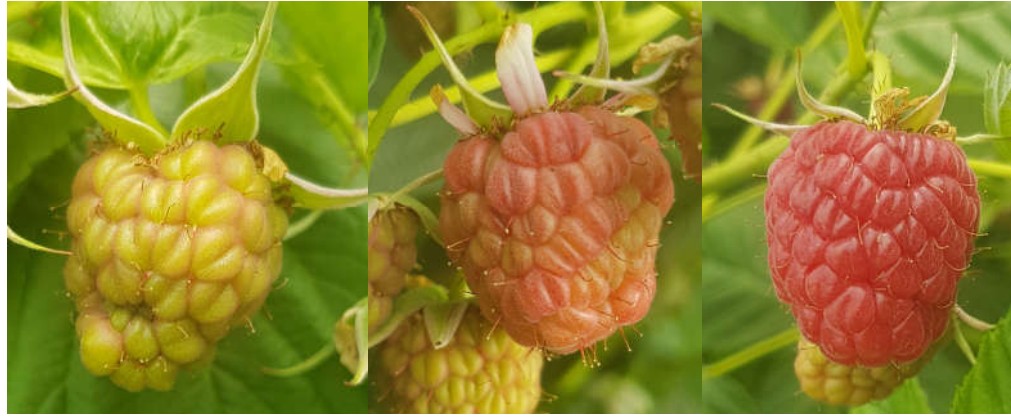

**Figure 1.** The three stages (green, ripening and ripe fruit) of raspberry fruit development inoculated in 2020 to investigate skin lesion development and stigmata infections by *Cladosporium*.

In 2019, for each fruit developmental stage, there was only one inoculation treatment without an un-inoculated control; fruits were inoculated with a mixture of three *C. cladosporioides* isolates. Fruits were inoculated on 29 August 2019, 5 September 19 and 12 September 2019. There were 12 pots (each with two canes) that were placed into four rows of three pots. Each row of plants was considered as one block since the three pots were fertigated by the same fertigation line. All 12 plants were inoculated on all three occasions—only a single branch was inoculated in each pot.

In 2020, there were four inoculation treatments: three isolates (as used in 2019) and a control. Inoculations were conducted on 9 September 2020, 16 September 2020 and 23 September 2020. There were 32 pots, each with three canes, in two rows. Each row of 16 pots was divided into four blocks, with one block containing four consecutive pots. One of the four pots in each block was randomly assigned to one of the four inoculation treatments. In each pot, 2–3 branches with the widest range of developmental stages were randomly selected for inoculation on each inoculation date.

### 2.2.2. Fruit Inoculation

*Cladosporium cladosporioides* isolates were grown on PDA (Oxoid, Basingstoke, UK, CM0139) for 7–10 days. Thirty millilitres of 0.5% KCl sterile water solution with 0.1% Tween-20 was placed onto the colony surface and agitated with a surface-sterilised glass rod, as recommended by Swett et al. [4]. The conidial suspension was decanted into a 25 mL universal bottle, and the concentration was determined using a haemocytometer and then adjusted to $10^7$ conidia/mL using sterile water. To make the mixed inoculum in 2019, the same volumes of the adjusted conidial suspensions for the three isolates were mixed together. For the control treatment in 2020, 0.5% KCl and 0.1% Tween-20 solution was used to inoculate fruits. Inoculations were carried out with a handheld sprayer to thoroughly wet individual fruit. Immediately after inoculation, each inoculated branch was covered with a polythene bag and plugged with a damp cotton bung. This was held in place with a cable tie to maintain high humidity inside the bag, which was left in place for 24 h and then removed. All the inoculated fruit on the branch were then removed and placed in separate trays.

In 2019, immediately after removal, fruits were surface-sterilised by gently washing the fruit in 0.1% Tween-20 for 2 min, 70% Ethanol for 30 s and, finally, 0.1% bleach for 2 min, then left in a laminar flow hood for approximately 30 min to dry, as described by Swett et al. [4], and placed in surface-sterilised trays for incubation at 20 °C in 24 h of light for four days before assessment.

In 2020, all the inoculated fruit on a single branch were removed into one punnet and then sterilised, as in 2019, but with an additional 2-min sterile distilled water wash at the end. Fruits were then placed into punnets, covered with a polythene bag to maintain high humidity and incubated at 20 °C in 24 h of light for four days.

### 2.2.3. Infection Assessment

In 2019, each individual fruit was assessed for its developmental stage and for the presence or absence of visual *Cladosporium* rot after incubation for four days. In 2020, each individual fruit was assessed for its developmental stage and the number of drupelets with visual *Cladosporium* symptoms on the following scale: 0—no drupelets infected, 1—one drupelet infected, 2—two drupelets infected, 3—three drupelets infected, 4—four drupelets infected, and 5—≥ 5 drupelets infected.

### 2.2.4. Statistical Analyses

To estimate the effect of inoculation date, fruit development stage and their interaction on the number (incidence) of *Cladosporium* infections for the 2019 inoculation, a Binomial generalised linear mixed model (GLMM) was fitted to the data using the glmmTMB package [15] in R version 4.0.3 (Vienna, Austria) [16]. In GLMM, residual errors were assumed to follow a binomial distribution, assuming no over-dispersion. Row (block),

date of inoculation and fruit development stage were added as fixed effect factors to the model. The plant (pot) was added as a random-effect factor to control for plant effects. An ordinal logistic regression model was fit to the 2020 skin lesion infection score data using the R "ordinal" package [17]. Five fixed-effect factors were used: block, inoculation date, *Cladosporium* isolate, *Cladosporium*-inoculated or not and fruit development stage. Fixed effects showed proportional odds, meeting the assumptions of ordinal regression. Green fruit (which all had no infection) were excluded from the analysis due to the complete separation causing converging issues with the model fit. The fitted ordinal model was used to generate estimated incidences of infections (either uninfected [score = 0] or infected [score ≥ 1]). The standard errors were also generated via the fitted ordinal model and adjusted by multiplying by the root of the estimated over-dispersion parameter. In all analyses, statistical significance was based on deviance analysis via a Chi-square test.

### 2.3. Susceptibility of Stigmata to Cladosporium Infection

### 2.3.1. Experimental Design

This experiment focused on stigmata infection in relation to fruit development stage (Figure 1) and surface-sterilisation, which was used to indicate if *Cladosporium* hyphae could penetrate stigmata or only colonise the stigmata surface. The same experimental planting was used as for the 2020 fruit inoculation experiment. Three inoculations were performed on: 9 September 2020, 16 September 2020 and 23 September 2020 (the number of raspberries assessed for each treatment are shown in Table S3). On each inoculation date, in addition to the fruit developmental stage and the four inoculation treatments (as for the skin lesion experiment), there was an additional treatment factor: with or without post-collection surface-sterilisation before incubation. Individual branches on each pot were randomly assigned to the "sterilised" or "unsterilised" treatment.

### 2.3.2. Fruit Inoculation

The same inoculation methods were used as described for the 2020 skin lesion experiments. All fruit on those selected branches were inoculated and picked 24 h after inoculation. When appropriate, fruit were surface-sterilised as described for the 2020 skin lesion susceptibility experiment; fruit from one branch were then placed into a single punnet covered with a polythene bag and incubated for a further four days at 20 °C before assessment.

### 2.3.3. Assessment

Individual fruit were scored for the severity of stigmata infections on the following scale: 0—no stigmata infected, 1—1–20% of stigmata infected, 2—21–40% of stigmata infected, 3—41–60% of stigmata infected, 4—61–80% of stigmata infected, and 5—81–100% of stigmata infected.

### 2.3.4. Statistical Analyses

An ordinal logistic regression was used to analyse the data as for the 2020 skin lesion data with one additional fixed-effect factor (sterilization vs. unsterilised). As all the green fruit scored 0, they were excluded from the analysis.

## 3. Results

### 3.1. Determination of Predominant Cladosporium Species on Raspberry Fruit

In total, molecular analyses of 54 isolates from raspberry fruit were performed, with 41 from the U.K. and 13 from Spain. Molecular analyses showed that in the U.K., the most frequently present *Cladosporium* species was *C. cladosporioides*, present on 41.5% of ripe fruit. The next most common species were *C. sphaerospermum* and *C. europaeum*, present on 14.6% of ripe fruit. On the U.K. fruit, 9.8% of isolates were identified as *C. limoniforme* and 7.3% as *C. ramotellum*. Finally, 12.2% of the isolates from U.K. grown raspberries gave a mixture of species from the BLAST database; however, their identities closely matched *C. fusiforme*.

Similarly, the predominant species present on fruit from Spain was *C. cladosporioides*, present on 84.6% of fruit. The only other species present was *C. sphaerospermum*, present on 15.4% of fruit (Table 1).

**Table 1.** The percentage breakdown of *Cladosporium* species present on raspberries collected from the U.K. and Spain. The total number of isolates was 54, with 41 from the U.K. and 13 from Spain.

| *Cladosporium* Species | U.K. Fruit | Spanish Fruit |
|---|---|---|
| *C. cladosporioides* | 41.5 | 84.6 |
| *C. fusiforme* | 12.2 | 0.0 |
| *C. sphaerospermum* | 14.6 | 15.4 |
| *C. europaeum* | 14.6 | 0.0 |
| *C. limoniforme* | 9.8 | 0.0 |
| *C. ramotellum* | 7.3 | 0.0 |

Phylogenetic analysis of both the ACT and TEF1$\alpha$ region corroborated with the species ID based on the nrBLAST database. Isolates showing close matches with *C. fusiforme* also closely matched a mixture of other *Cladosporium* species (however, with poor query coverage), which were also added to the phylogenetic analysis. These isolates, however, still formed a clade with the *C. fusiforme* reference sequences in the phylogenetic trees.

In both trees (Figures 2 and S1), a mixture of *Cladosporium* isolates were found in individual farms; but for those isolates closely related to *C. fusiforme*, three of the five isolates originated from one farm in Kent.

### 3.2. Fruit Susceptibility to Cladosporium Skin Lesions

In 2019, *Cladosporium* skin lesions were only present on fruit at the ripening and ripe stage. The overall incidence of fruit with *Cladosporium* skin lesions were 13.2% (s.e. $\pm$ 8.3%) on ripening fruit and 26.5% (s.e. $\pm$ 11.5%) on ripe fruit. There was no significant difference in the presence of *Cladosporium* on ripening and ripe fruit ($X^2$ (1, $n = 36$) = 1.06, $p = 0.303$).

In 2020, *Cladosporium* skin lesions were present only on fruit at the ripening and ripe stages, with the overall incidence being 50.2% (s.e. $\pm$ 12.1%) of ripe fruit with skin lesions and 17.4% (s.e. $\pm$ 9.0%) of ripening fruit with skin lesions. Ripe fruit were more susceptible to *Cladosporium* than ripening fruit: the odds of fruit with *Cladosporium* skin lesions at a given severity score (odds = infected/healthy) increased to 5.14 (s.e. $\pm$ 1.3) ($p < 0.001$) times when fruit moved from the ripening to ripe stage. Inoculation led to more fruit with skin lesions with an overall incidence of *Cladosporium* of 36% (s.e. $\pm$ 9.9%) in inoculated and 20.4% (s.e. $\pm$ 13.2%) in the control ($p < 0.05$). There was also a significant difference in *Cladosporium* skin lesion scores amongst the three isolates ($p < 0.05$): isolate three caused significantly higher skin lesion severity scores when compared to isolate two.

### 3.3. Infection of Stigmata by Cladosporium

Stigmata infections occurred across all fruit developmental stages irrespective of surface-sterilisation (Figure 3). There were significant differences in the severity scores between inoculation dates ($p < 0.001$), inoculated or uninoculated ($p < 0.001$), three *Cladosporium* isolates ($p < 0.001$) and surface-sterilisation ($p < 0.001$) on the severity of stigmata infections. There were no significant differences in stigmata infection scores across developmental stages ($p = 0.166$).

There were higher incidences of *Cladosporium* on stigmata across all the developmental stages in the unsterilised treatments compared to the sterilised treatments (Table 2).

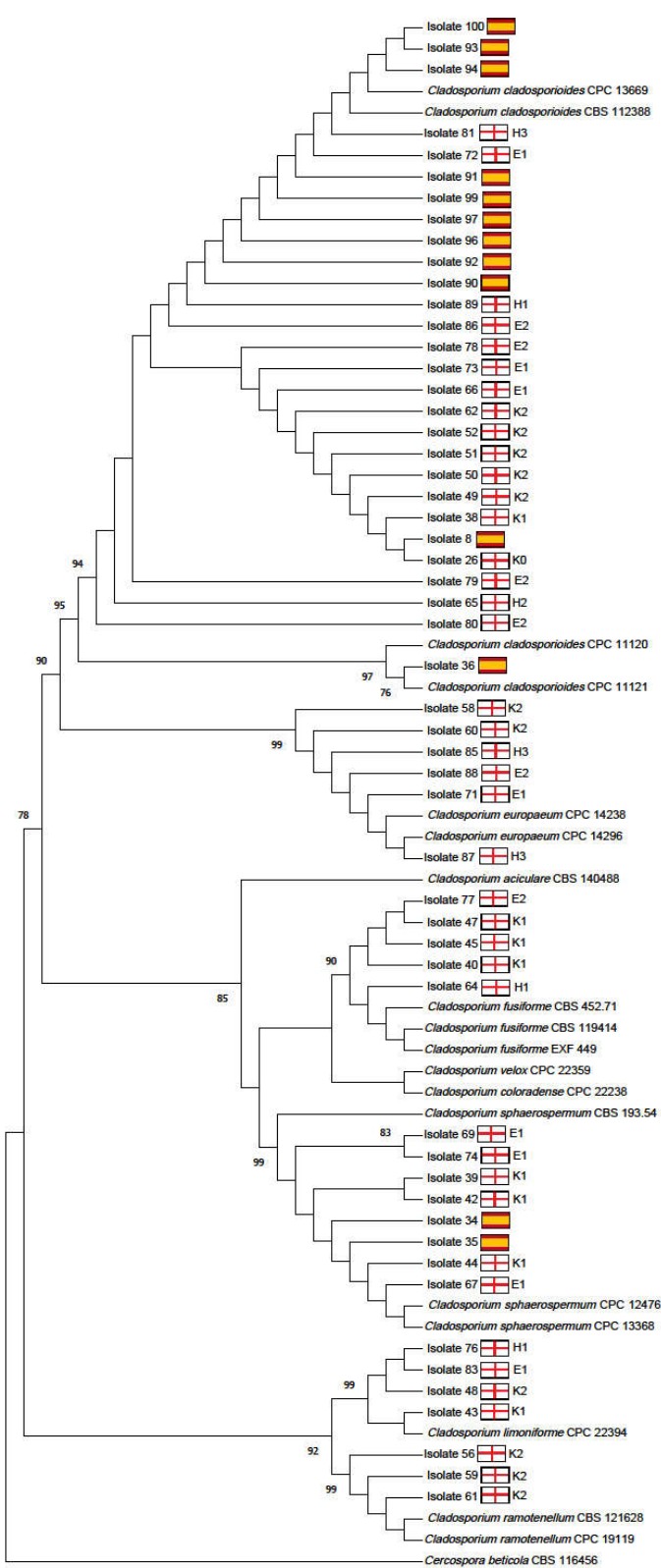

**Figure 2.** A phylogenetic tree of the ACT gene for 72 isolates of *Cladosporium*. The tree was created using maximum likelihood with a Kimura 2-parameter evolutionary model and 1000 bootstrap replications. The numbers above branches represent bootstrap values. The flags represent the country of origin, the codes represent areas within the country and numbers denote different farms. Area codes are as follows: E = Essex, K = Kent, H = Herefordshire.

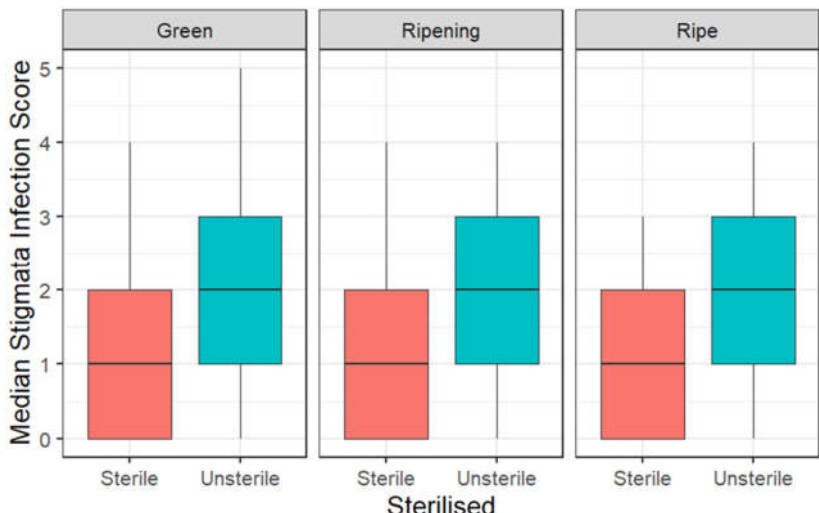

**Figure 3.** A boxplot of stigmata infection scores across the three developmental stages and between sterile vs. unsterile raspberry fruits; the black horizontal line is the median score.

**Table 2.** The overall incidence (% fruit with *Cladosporium* present) of *Cladosporium* on stigmata across four raspberry developmental stages and between sterilisation treatments. s.e.: standard error of the means.

| Sterilisation Treatment | Green | Ripening | Ripe |
|---|---|---|---|
| Sterilised | 66.4 (s.e. $\pm$ 6.9) | 74.5 (s.e. $\pm$ 12.3) | 69.9 (s.e. $\pm$ 16.5) |
| Unsterilised | 88.3 (s.e. $\pm$ 6.8) | 87.0 (s.e. $\pm$ 11.1) | 90.6 (s.e. $\pm$ 8.6) |

The incidences of *Cladosporium* infections on stigmata were also higher across all developmental stages inoculated with isolate three vs. isolate one and two (Table 3).

**Table 3.** The overall incidence (% fruit with *Cladosporium* present) of *Cladosporium* on stigmata across raspberry developmental stages and between inoculums. s.e.: standard error of the means.

| Inoculum | Green | Ripening | Ripe |
|---|---|---|---|
| Control | 65.7 (s.e. $\pm$ 10.5) | 68.7 (s.e. $\pm$ 14.3) | 70.1 (s.e. $\pm$ 14.7) |
| Isolate 1 | 79.1 (s.e. $\pm$ 9.1) | 81.2 (s.e. $\pm$ 10.7) | 82.2 (s.e. $\pm$ 10.8) |
| Isolate 2 | 81.0 (s.e. $\pm$ 8.7) | 82.9 (s.e. $\pm$ 10.1) | 83.9 (s.e. $\pm$ 10.7) |
| Isolate 3 | 87.9 (s.e. $\pm$ 6.4) | 89.3 (s.e. $\pm$ 7.3) | 89.9 (s.e. $\pm$ 7.2) |

## 4. Discussion

This study has shown that *C. cladosporioides* is the predominant species infecting raspberry fruit in the U.K. and Spain, consistent with findings in the U.S.A. [4]. Several other *Cladosporium* species were also found on ripe raspberries, including *C. sphaerospermum*, *C. europaeum*, *C. limoniforme*, *C. ramotellum* and *C. fusiforme*. Of particular interest is the first isolation of *C. fusiforme* from plant material, suggesting the possibility of its presence on other crops. Three of the five isolates of this species were detected at one farm in Kent, indicating that this farm may provide an ecological niche favourable to this species. Some of these *Cladosporium* species were tested for their pathogenicity on other important horticultural crops, such as *C. limoniforme* on strawberries [18] and *C. cladosporioides*, *C. limoniforme* and *C. ramotellum* on pears and apples [19].

As *C. cladosporioides* was the most prevalent species found on raspberries, a large proportion of the primary inoculum may arise from airborne spores, as well as from crop debris [20], as this species comprises a large proportion of the airborne fungal load [21]. Previous studies investigating the patterns of *Cladosporium* airborne spores have focused on

indoor and outdoor environments [22,23]. It may, therefore, be necessary to investigate how agricultural environments, e.g., inside polytunnels, impact the inoculum load and, hence, disease risk. There also appear to be differences among the three *C. cladosporioides* isolates in causing skin lesions and stigmata infections. Thus, in addition to inter-species differences in pathogenicity, further work is also needed to assess the intra-species variability in pathogenicity on raspberry fruit.

Present results indicate that ripe raspberries are most susceptible to *Cladosporium* skin lesions, followed by ripening fruit. In contrast, green fruit are not susceptible to *Cladosporium* skin lesions. Similarly, Swett et al. [4] showed that ripe raspberries are most susceptible to *Cladosporium* infection. Due to ripe fruit being delicate and susceptible to attacks by pests such as *Drosophila suzukii*, wounds may provide entry points for *Cladosporium* species, as found in the U.S.A. [4]. In addition, manual fruit harvesting may cause small abrasions to the surface, allowing access for fungi such as *Cladosporium* to infect and cause visible symptoms post-harvest.

Fruit from the green developmental stage onwards appear susceptible to *Cladosporium* stigmata infections. The severity of stigmata infections decreased when the fruit were surface-sterilised, indicating a significant proportion of *C. cladosporioides* inoculum only colonised the stigmata surface. Thus, the stigmata of raspberries are susceptible to *C. cladosporioides* and this species grows more readily as a saprophyte on the stigmata surface. The present results are consistent with a previous finding that *Cladosporium* rarely invaded the internal tissues of the grape fruiting structure [24]. *Cladosporium cladosporioides* can colonise or infect raspberry stigmata early in fruit development. However, further research is needed to study the dynamics of *Cladosporium* spore survival on the fruit surface. It is unknown if *Cladosporium* spores could survive long enough to cause secondary infections from the stigmata onto the skin's surface. Similar secondary infections were observed with *Botrytis cinerea* on raspberries [25]. *Cladosporium cladosporioides* was demonstrated to infect strawberry flowers [8]; therefore, similar investigations into the susceptibility of raspberry flowers to *Cladosporium* infections are needed.

Overall, it may be more economical to focus on managing *Cladosporium* infection when fruit are most susceptible to skin lesions, especially at the ripening stages of development. Improved husbandry and hygiene by removing decaying material may aid in reducing the inoculum load to prevent both skin and stigmata infections. The use of chemical fungicides to manage *Cladosporium* is limited unless there are effective products with a short half-life. As *Cladosporium* mainly grows on the stigmata surface, biological control agents may be able to compete effectively with *Cladosporium*, leading to much reduced incidences of raspberries with *Cladosporium* infections. The use of biocontrol agents against the *Cladosporium* species was demonstrated in multiple studies. Examples include multiple species of *Trichoderma* inhibiting the growth of *C. herbarum* [26] and *Chaetomium globosum* inhibiting the growth of multiple isolates of *C. cladosporioides* [27] during in vitro plate assays. Further research is needed to demonstrate the efficacy of such biological control agents in field conditions against the *Cladosporium* species.

Overall, the data obtained in this study will be beneficial in understanding the predominant *Cladosporium* species impacting raspberries and the growth stages of the ripening fruit most susceptible to *Cladosporium* infections. This will benefit agronomists and growers in more accurately assessing *Cladosporium* rot symptoms and identifying the most appropriate, and timing of, the application of control measures.

**Supplementary Materials:** The following supporting information can be downloaded at: https://www.mdpi.com/article/10.3390/horticulturae9020128/s1, Figure S1: A phylogenetic tree of the TEF1α gene for 70 isolates of *Cladosporium*; Table S1: The sequences of the ACT and TEF1α primers; Table S2: The accession numbers of the reference sequences used in the phylogenetic analyses; Table S3: The number of raspberries inoculated and sterilised at each inoculation date.

**Author Contributions:** Conceptualization, L.H.F., X.X. and N.M.; methodology, L.H.F., X.X., N.M., A.L.H. and T.P.; formal analysis, L.H.F., X.X., C.V.-V. and G.D.; investigation, L.H.F., A.L.H. and G.F.; resources, L.H.F., T.P. and G.D.; software, L.H.F., X.X. and G.D.; validation, L.H.F., X.X. and G.D.; writing—original draft preparation, L.H.F.; writing—review and editing, X.X., N.M. and G.D.; visualization, L.H.F. and G.D.; supervision, X.X. and N.M.; project administration, L.H.F.; funding acquisition, X.X. and N.M. All authors have read and agreed to the published version of the manuscript.

**Funding:** This research is funded by the Biotechnology and Biological Sciences Research Council (BBSRC) Collaborative Training Partnerships (CTP) for Fruit Crop Research in partnership with NIAB and Cranfield University, BBSRC (BB/T509073/1) and an industry partner with Berry Gardens Ltd.

**Institutional Review Board Statement:** Not applicable.

**Informed Consent Statement:** Not applicable.

**Data Availability Statement:** The data generated and analysed in this study are available on request from the corresponding author.

**Acknowledgments:** We would like to thank Sarah Cohen for her initial support in the DNA amplification and sequencing. We would also like to thank Richard Harnden and Andrius Kumstys (Berry Gardens Ltd.) for their support in collecting raspberry samples from across the U.K. Finally, we would like to thank Harriet Duncalfe for her advice and guidance during the experiments.

**Conflicts of Interest:** The authors declare no conflict of interest.

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
