# Peer review of "Cladosporium Species: The Predominant Species Present on Raspberries from the U.K. and Spain and Their Ability to Cause Skin and Stigmata Infections"

_horticulturae, doi:10.3390/horticulturae9020128_

Round 1

Reviewer 1 Report

The current study reports an interesting topic that points out to analyse the diversity of Cladosporium Species to identify The Predominant Species Present on U.K. Raspberries and their Ability to Cause Skin and Stigmata Infections. The manuscript shows originality and novelty, but major needed adjustments in its standard English. Therefore, I ask the authors to pass their manuscript to a native English speaker for editing and revision. The presented parts are significant and interpreted appropriately. The raised conclusions and further suggestions are justified. The study covers its topic which is well relevant, and all used references are appropriate. Also, the study is correctly designed and sounds technically. Just pay attention to the citation style along the manuscript. Title need to indicate Author of the species and also in parenthesis Order: Family. The Abstract part is well written, aiming and clear. Just kindly add some numbers to this part as it sympathetically enriches it. On the other hand, all keywords fit well but you need more, when you add them, please ensure that these new words are not in the title. The Introduction part is well structured and aiming. It needs some adjustments in terms of linguistic mistakes and sentences reformulation in the impersonal form rather than the first voice’s one. I also want you to add a new paragraph where you clarify the importance of your study, for instance provide a real problem, a justification, and the benefits that this study will provide to agricultural ministry. This is because in line 51 your justification is that topic is “still unknown” and that is the reason of science, so it is not a justification. The Materials and methods part is well structured and clear. All adopted methods and materials were appropriately described and mentioned and are well clear for further repetition by other researchers. You need to add a new section where you explain clearly “Study areas” and “Field experiments design”. In this section state clearly the kind of study of this investigation and the criteria for selecting fruits, or experimental units and sampling units. Please add a new sentences to explain raspberry variety or c.v. identification, I understand that it was done for performing comparisons, but you need to state clearly in the manuscript your reasons and they must be supported by references. Only minor linguistic adjustments are requested within this part. In data analysis you used several univariate statistical methods, however there is no justification of their use. Please state in that section which are the assumption or characteristics of data that allowed you to use them. The Results part is clear and well aiming. Some adjustments are only needed in terms of sentences reformulation in a better standard language or in a less cumbersomeness manner. Other minor linguistic mistakes should be also adjusted. Although, the scientific analysis of the findings is well performed and the correct statistical approach was performed adequatel.The Discussion part is well structured and aiming. It needs adjustments related to linguistic mistakes, sentences reformulation in the impersonal form rather than the first voice’s one, sentences reformulation in a better standard English, a more concise and less cumbersomeness manner. Also, some statements lack reliable sources (references) that should be provided. However, all mentioned sources are reliable and directly related to the study’s findings. The Conclusions part is very well structured and aiming. It summarizes appropriately the findings of the currents study and suggests further related research. Minor adjustments related to sentence reformulation in a less cumbersome manner and other linguistic mistakes are only needed herein. There are some suggestions in the attachment.

Briefly, based on the above and below detailed explanation, the manuscript needs importante adjustments, and at this moment is required to amend it to have a future merit to be published in “Horticulturae” journal once all suggestions and recommendations are fully addressed.

Author Response

Dear Reviewer One,

We, the authors, want to thank you for taking the time to review our paper and give us constructive feedback. To make it easier for you to see what we have edited from your suggestions, below we have put your comments in blue, and our answer below in black.

Just pay attention to the citation style along the manuscript

Answer: We have edited all citations to follow the journal guidelines.

Title need to indicate Author of the species and also in parenthesis Order: Family.

Answer: We have added order and family to the Cladosporium species mentioned in the introduction of the paper.

(Abstract) Just kindly add some numbers to this part as it sympathetically enriches it.

Answer: Due to the word limitations, we could not add more detailed statistics in the abstract. We have added the percentage of the predominant species of Cladosporium.

On the other hand, all keywords fit well but you need more, when you add them, please ensure that these new words are not in the title

Answer: The keyword Cladosporium has been removed, and the new keyword Ordinal Regression has been added.

(Introduction) I also want you to add a new paragraph where you clarify the importance of your study, for instance provide a real problem, a justification, and the benefits that this study will provide to agricultural ministry. This is because in line 51 your justification is that topic is “still unknown” and that is the reason of science, so it is not a justification.

Answer: We have added a paragraph towards the end of the introduction justifying the study.

(Methods) You need to add a new section where you explain clearly “Study areas” and “Field experiments design”. In this section state clearly the kind of study of this investigation and the criteria for selecting fruits, or experimental units and sampling units.

Answer: We have now re-organised and added more detail to the sections titled experimental design to make it clearer what the block organisation and fruit selection criteria.

Please add a new sentences to explain raspberry variety or c.v. identification, I understand that it was done for performing comparisons, but you need to state clearly in the manuscript your reasons and they must be supported by references.

Answer: The variety selected for use in the field experiments is a popular commercial cultivar used by U.K. growers where Cladosporium has been previously reported on in personal communications with growers and agronomists. We have included a short sentence to reflect this in the text.

In data analysis you used several univariate statistical methods, however there is no justification of their use. Please state in that section which are the assumption or characteristics of data that allowed you to use them.

Answer: Our statistician has informed us these are all the expected analyses given the data types. We’ve updated the relevant parts of section 2.2.4. to make this clear.

(Discussion) Also, some statements lack reliable sources (references) that should be provided.

Answer: We have edited the discussion to make it clearer that there are areas within the literature that are lacking and need further research. Other statements have had references added.

Multiple comments on the English throughout the paper

Answer: The paper has been edited to remove linguistic mistakes in all sections and reviewed by a native English speaker.

Suggestions made via PDF document

Clarifying if strains or isolates investigated

Answer: We have now clarified we were investigating isolates, not strains.

Skin lesions in fruits?

Answer: We have now clarified this is referring to skin lesions on the fruit surface.

Dollars instead of pounds

Answer: The referenced amount was from the government body DEFRA who only publish their findings in U.K. pounds. Hence, we have included both pounds and American dollars, which was calculated using the average conversion rate of 2020.

References [2],[3], Revise

Answer: We have updated the references to better reflect the statements made.

Introduction, lines 49 and 50

Answer: The authors of these species have been included.

Did we consider variety?

Answer: We did take note of all the varieties the isolates were collected from. In the U.K. all isolates were collected from Driscolls Maravilla, apart from one which originated from a breeding line plant (and hence isn’t an official variety). The Spanish isolates were from a mix of Wengi, Malling Bella and Adelita. This information can be provided in a supplementary table if requested.

Adding a map

Answer: We have not included a map of the farms as the individual growers have not given us permission to publicise where we sourced our fruit from. We have included codes for the counties the fruits were collected from in the U.K., which we hope will suffice.

Please indicate what type of experiment design was followed

Answer: As before, this section of the paper has been re-written to make it clearer.

Co-ordinates/Map

Answer: We have now added the co-ordinates of the polytunnel used.

Once again, thank you for taking the time to make these suggestions, and we hope we have satisfactorily edited the paper as such.

Yours faithfully,

Lauren Farwell

Reviewer 2 Report

The paper is well organized with some necessary corrections listed below.

26- Keywords are those which do not appear in your title

37- suzukii

281- I think you may have a problem with the description of your tree figure. You have some right-justified text here, and continues in a normal justified fashion on the next page. There is no title for this figure?

315- You should leave a space after tables prior to adding a new sentence. It makes your text look like a footer to the table, rather than an independent line of text. Also, please check the style guide for the journal; your tables have too many internal lines.

325- …this species were detected…

340- …polytunnels, impact the inoculum…

358- This theory needs to be backed up by further review evidence showing the longevity of the Cladosporium spores on fruit. Can they really persist on a fruit surface from green to ripe stage?

400- Your references follow a variety of patterns. Please check the style guide and apply the format uniformly to each reference.

Author Response

Dear Reviewer Two,

We, the authors, want to thank you for taking the time to review our paper and give us constructive feedback. To make it easier for you to see what we have edited from your suggestions, below we have put your comments in blue, and our answer below in black.

26- Keywords are those which do not appear in your title

Answer: All words contained in the title have now been removed.

37- suzukii

Answer: Has been edited to the correct name.

281- I think you may have a problem with the description of your tree figure. You have some right-justified text here, and continues in a normal justified fashion on the next page. There is no title for this figure?

Answer: The text in the figure legend has been re-adjusted and is now in full view.

315- You should leave a space after tables prior to adding a new sentence. It makes your text look like a footer to the table, rather than an independent line of text. Also, please check the style guide for the journal; your tables have too many internal lines.

Answer: A line space has been left after every table to make the text and tables clearer. The extra lines within the table have been removed to match the journal style guide.

325- …this species were detected…

Answer: Has been edited to “were”.

340- …polytunnels, impact the inoculum…

Answer: Edited to remove “on”.

358- This theory needs to be backed up by further review evidence showing the longevity of the Cladosporium spores on fruit. Can they really persist on a fruit surface from green to ripe stage?

Answer: We have searched the literature and found no studies that have investigated the long-term viability of Cladosporium spores. We have now edited the text to reflect this is unknown in the literature and needs further investigation.

400- Your references follow a variety of patterns. Please check the style guide and apply the format uniformly to each reference.

Answer: The references have been edited to follow the journal format.

Once again, thank you for taking the time to make these suggestions, and we hope we have satisfactorily edited the paper as such.

Yours faithfully,

Lauren Farwell

Reviewer 3 Report

The manuscript presents results of field survey of Cladosporium species (identified using molecular approaches) on raspberry fruits collected in UK and Spain and results of innoculation experiments to reveal among others the effect of fruit development stage on susceptibility to infection and test if stigmata can be infected, too. Overall, the manuscript is well written, introduction provides logic overview of problem and aim of study is well defined. Materials and methods section describes all important details including sound statistical analysis of data. Results are well presented and for sure in sufficient amount for Commnications (I would even say that they are enough for Article). Discussion also covers all important findings which are compared to current knowledge. I have only few suggestions for minor revision below.

Specific comments:

Title - since there are also interesting data from Spain, title modification to: "... on U.K. and Spain Raspberries ..." might be more appropriate and Spain also mentioned in abstract (e.g. "In comparison, results from Spain show ...")

line 45 - instead of (2019) numerical format of citation should be used

line 49 and other occurrence - Usually when species name starts the sentence genera name is not abbreviated.

line 65 - please write manufacturer name, city and country in all products used

line 141 - Fruits were ...

line 147 - please specify humidity and light conditions if it is appropriate

Figure 1 - quality (focus/resolution) does not seem to be very high, also photos could be enlarged

Results

Some comments on data from Spain should be added into text with reference to table 1

line 281-282 there is some problem with missing text (figure overlaps text?)

line 373 - can you add some example of some promising biocontrol agent, e.g. Trichoderma with reference/s?

Author Response

Dear Reviewer Three,

We, the authors, want to thank you for taking the time to review our paper and give us constructive feedback. To make it easier for you to see what we have edited from your suggestions, below we have put your comments in blue, and our answer below in black.

Title - since there are also interesting data from Spain, title modification to: "... on U.K. and Spain Raspberries ..." might be more appropriate and Spain also mentioned in abstract (e.g. "In comparison, results from Spain show ...")

Answer: Included Spain in the title to reflect those results. Due to the word limitations, we could not add a more detailed description of the species breakdown in the abstract, but have reflected that the predominant species on Spanish fruit was the same as fruit from the U.K.

line 45 - instead of (2019) numerical format of citation should be used

Answer: The numerical format has been edited to the citation format.

line 49 and other occurrence - Usually when species name starts the sentence genera name is not abbreviated.

Answer:  All occasions of the genera name beginning in a sentence has now been edited.

line 65 - please write manufacturer name, city and country in all products used

Answer: Edited to include the above details to products used.

line 141 - Fruits were ...

Answer: Edited “fruit” to “fruits”.

line 147 - please specify humidity and light conditions if it is appropriate

Answer: As the raspberry fruit are respiring in the punnets we would expect them to be at >98% RH during incubation. As the punnets were individually covered in polythene a high humidity would have been maintained during the 4 day period (we have now added the polythene bags were used to maintain a high humidity and the light conditions used).

Figure 1 - quality (focus/resolution) does not seem to be very high, also photos could be enlarged

Answer: Photos have been replaced with better quality images and the images have been enlarged.

Some comments on data from Spain should be added into text with reference to table 1

Answer: A few sentences have been added regarding the isolates from Spain in reference to table 1.

line 281-282 there is some problem with missing text (figure overlaps text?)

Answer: The text for the figure legend has been re-adjusted and is now visible.

line 373 - can you add some example of some promising biocontrol agent, e.g. Trichoderma with reference/s?

Answer: Have added examples of biocontrol agents demonstrated to work against Cladosporium species in vitro. We hope this additional information is satisfactory.

Once again, thank you for taking the time to make these suggestions, and we hope we have satisfactorily edited the paper as such.

Yours faithfully,

Lauren Farwell

Round 2

Reviewer 1 Report

I see a new revised version of the ms. I have no more comments. 

Congrats.